# Diagnosis of *Acanthamoeba* Keratitis: Past, Present and Future

**DOI:** 10.3390/diagnostics13162655

**Published:** 2023-08-11

**Authors:** Matthew Azzopardi, Yu Jeat Chong, Benjamin Ng, Alberto Recchioni, Abison Logeswaran, Darren S. J. Ting

**Affiliations:** 1Department of Ophthalmology, Royal London Hospital, London E1 1BB, UK; matthew.azzopardi.14@um.edu.mt; 2Birmingham and Midland Eye Centre, Birmingham B18 7QH, UK; benjamin.ng1@nhs.net (B.N.); alberto.recchioni@nhs.net (A.R.); 3Academic Unit of Ophthalmology, Institute of Inflammation and Ageing, University of Birmingham, Birmingham B15 2TT, UK; 4Moorfields Eye Hospital, London EC1V 2PD, UK; abison.logeswaran.1@city.ac.uk; 5Academic Ophthalmology, School of Medicine, University of Nottingham, Nottingham NG7 2RD, UK

**Keywords:** *Acanthamoeba* keratitis, diagnosis, culture and microscopy, corneal biopsy, in vivo confocal microscopy, polymerase chain reaction, anterior segment optical coherence tomography, next-generation sequencing, artificial intelligence

## Abstract

*Acanthamoeba* keratitis (AK) is a painful and sight-threatening parasitic corneal infection. In recent years, the incidence of AK has increased. Timely and accurate diagnosis is crucial during the management of AK, as delayed diagnosis often results in poor clinical outcomes. Currently, AK diagnosis is primarily achieved through a combination of clinical suspicion, microbiological investigations and corneal imaging. Historically, corneal scraping for microbiological culture has been considered to be the gold standard. Despite its technical ease, accessibility and cost-effectiveness, the long diagnostic turnaround time and variably low sensitivity of microbiological culture limit its use as a sole diagnostic test for AK in clinical practice. In this review, we aim to provide a comprehensive overview of the diagnostic modalities that are currently used to diagnose AK, including microscopy with staining, culture, corneal biopsy, in vivo confocal microscopy, polymerase chain reaction and anterior segment optical coherence tomography. We also highlight emerging techniques, such as next-generation sequencing and artificial intelligence-assisted models, which have the potential to transform the diagnostic landscape of AK.

## 1. Introduction

*Acanthamoeba* keratitis (AK) is a severe sight-threatening corneal infection. It was first reported in 1974 by Naginton et al., who described it as a rare form of infectious keratitis post-ocular trauma [1]. Being an opportunistic protozoan, *Acanthamoeba* can be found in many different environmental niches, including reservoirs, sea water, natural thermal water, soil, dust, and healthy nasal mucosa [2,3,4,5,6,7,8].

Various *Acanthamoeba* species exist, and these species are largely classified either on the basis of their morphological characteristics and cyst size, as suggested by Pussard and Pons [9], or their 18S ribosomal RNA gene sequencing [10]. Using the latter classification approach, 23 genotypes have been reported (T1-T23), with the T4 genotype being the most common causative agent of AK [11]. Other genotypes reported to cause AK include T2, T5, T6, T10, T11, T12 and T15 [12,13,14,15,16,17].

The global annual incidence of AK has been estimated to be 2.9 cases per million people per year [18], with significant geographical variations among different countries [19]. The highest estimated incidence rate occurs in India, with 15.2 cases per million people, whilst in Europe, the United Kingdom (UK) has the highest incidence rate at 4.3–14.6 cases per million per year [18,20]. Epidemiological studies have shown that AK is on the rise, with the number of AK cases annually diagnosed at Moorfields Eye Hospital being three times higher between 2011 and 2014 than between 2004 and 2010 [21]. Similar findings were identified in studies from the United States [22,23] and Australia [24]. This increase in AK prevalence is thought to be related to the increased use of contact lenses, which is known to be the main risk factor in developed countries [25,26,27,28,29,30,31,32,33].

The diagnosis of AK can be challenging due to its initial non-specific presentation. It is typically unilateral, though it can be bilateral [34,35]. Typical initial clinical features are pain, photophobia, watering and red eye, as well as epithelial irregularities, opacities, microerosions, microcystic oedema, and patchy stromal infiltrates [36,37,38]. The degree of pain is classically described to be severe and disproportionate to the clinical picture [32,39]. However, AK can also be painless, leading to diagnostic uncertainties [40,41,42]. Dendritiform epitheliopathy resembling *Herpes simplex* keratitis (HSK), either with or without stromal and endothelial inflammation, often develops in the early stages of AK cases [43]. Due to this fact and the initial non-specific acute presentation, AK is often initially misdiagnosed as adenoviral keratitis or HSK [32,34,41,44,45,46]. 

Another early sign that is highly suggestive of AK is radial keratoneuritis [47], which is thought to be the cause of the severe disproportionate pain. Even though it is often considered to be pathognomonic for AK, it can also occur in bacterial keratitis, especially *Pseudomonas* keratitis [48,49]. This outcome is often seen early in the disease process, and it becomes harder to identify as the infection worsens. Finally, other signs that can be seen in AK include immune ring infiltrates and scleritis due to stromal progression [40,50,51,52], as well as limbal hyperaemia and oedema [53,54,55]. 

All of this information illustrates the difficulty with early differentiation of AK from other causes of keratitis. However, timely and accurate diagnosis is crucial during the management of AK, as delayed diagnosis has been shown to increase the risk of treatment failure with poor visual outcomes [21,45,56,57]. 

Currently, AK diagnosis is primarily achieved through a combination of clinical suspicion, microbiological investigations and corneal imaging [58]. The presence of suspicious clinical symptoms and signs, along with risk factors, such as contact lens wear and corneal trauma, should raise suspicion of AK [59,60]. When considering appropriate diagnostic techniques, having a test with high sensitivity (i.e., low false negative results) and high specificity (i.e., low false positive results) is ideal, particularly in the management of AK. A false negative result can lead to missed AK, which results in a delay in treatment and worse prognosis, whereas a false positive result can lead to incorrect diagnosis and unnecessary or inappropriate administration of anti-acanthamoebic drugs that are potentially toxic to the ocular surface and the eye [21,61]. In view of the expanding evidence in the literature, this review aimed to examine the diagnostic performance of various techniques and tools and discuss the potential role of artificial intelligence (AI) in improving the diagnosis of AK.

## 2. Culture and Microscopy

Historically, corneal scraping for microbiological culture has been considered to be the gold standard for diagnosis of AK. Non-nutrient agar plates seeded with *Escherichia coli* are the medium most commonly used to culture *Acanthamoeba* [62,63,64], with *E. coli* acting as a food source for the trophozoites [43,62,63,64]. Typically, migratory tracks with trophozoites at each end are observed. Whilst culture is easy to perform and is cost-effective, a number of issues limit its use as a robust sole diagnostic test in AK. Firstly, a positive culture result may require as long as 10 days of incubation [43], which can delay the diagnosis and treatment [64]. Furthermore, whilst microbiological culture has a high specificity (100%), its sensitivity is reported to range from 33.3 to 66.7% depending on the technique used [64,65,66,67], which can risk doctors missing the diagnosis of AK. In negative corneal cultures, culture of the CL solution from the storage case might be useful [56], though positive results might simply represent poor CL hygiene and should be interpreted with caution.

An important consideration related to the sampling technique required in order to improve the yield rate is obtaining corneal samples for microbiological investigations before initiation of any antimicrobial treatments [68]. Some other commonly used drops have also been shown to have a deleterious effect on the in vitro survival of *Acanthamoeba*. For example, topical anaesthesia, such as proxymetacaine 0.5%, oxybuprocaine 0.4% and tetracaine 1%; the drop-preservative benzalkonium chloride (commonly found in preserved-levofloxacin drops started for keratitis); and povidone iodine have all been shown to be toxic to *Acanthamoeba* trophozoites and cysts [69,70]. Furthermore, common dry eye drops, such as Systane Ultra, have been shown to have some anti-*Acanthamoeba* properties in vitro, which occur due to the presence of 0.4% polyethylene glycol 400 [71]. All of this information can potentially affect the viability of *Acanthamoeba*, increasing false-negative results. However, real-world studies are required to assess the clinical significance of these in vitro findings. The culture yield can be further affected via use of the scraping technique and the choice of instruments. In infectious keratitis, it has been reported that corneal sampling with cotton-tipped applicators provides a higher rate of positive cultures than samples taken using a blade [72]. This outcome is, however, different for AK corneal sampling, with modified bezel needles shown to be superior to both scraping using a blade and cotton swabs [73].

Direct microscopy and staining are also used for corneal scrapes. A number of staining techniques have been found to aid the visualisation of trophozoites and cysts. In most cases, cysts exhibit features such as double-walled structures, with the endocyst (inner wall) visualised as a distinct layer from the ectocyst (outer wall) (Figure 1). The shape of the endocyst can be stellated, polygonal, round or oval; the ectocyst has a variably stained background [74,75]. However, these cysts may occasionally be mistaken for other commoner cells, such as macrophages and mononuclear cells, especially whilst performing Gram and Giemsa staining [76].

Well-documented examples include calcofluor white (CFW) and potassium hydroxide (KOH) wet mount. CFW is a chemofluorescent dye with an affinity for the polysaccharide polymers of *Acanthamoeba* cysts, including cellulose and chitin [75]. Studies have claimed that this approach is a simple, rapid and highly reliable staining technique used to detect amoebic cysts [75,78,79]. However, even though positive CFW tests are available on the same day and have a high specificity of 96%, their sensitivity has been reported to be 71%, making a negative CFW insufficient to rule out AK [78]. On the other hand, KOH wet mount has been reported to have a sensitivity and specificity of 91.4% and 100%, respectively, for *Acanthamoeba* [80].

Other stains that have been used to detect trophozoites and cysts include iodine, haematoxylin and eosin (H&E); Gömöri methanamine silver (GMS); periodic acid-Schiff (PAS) stains; and Gimenez stains [74,79,81]. Lactophenol cotton blue can also be used to stain *Acanthamoeba* cysts [82], whilst fluorescein-conjugated lectins can be used to stain both cysts and trophozoites [83]. Another useful stain is Kop-color^®^ (Fumouze Diagnostics, Levallois-Perret, France), which has been shown to stain *Acanthamoeba* cysts into a yellow–orange colour on a blue–purple background, providing good contrast [84]. Stains such as Giemsa, Gram and acridine orange are reported to be the least useful [74,79,81,85]. 

All of the above evidence suggests that a combination of stains is superior to a single staining technique in diagnosing AK, and it should be an adjunct to other diagnostic tests such as culture.

## 3. Corneal Biopsy

In cases in which the initial culture and microscopy of corneal scrapes is negative, histological analysis, immunofluorescence and culture of corneal stromal biopsy (with microscopy and staining) are the other diagnostic techniques that can be considered [51,86,87,88,89]. A number of reports in the literature describe positive corneal biopsies following negative scrapes [87,88,90,91,92], with the reported sensitivity of stromal biopsy histological analysis for AK being 65% [93]. This result is thought to reflect the greater amount and depth of tissue obtained via biopsy, as with AK progression, *Acanthamoeba* penetrates deeper into the stroma, making it harder for corneal scrapes to pick it up.

There are a number of corneal biopsy techniques in clinical use [94]. Depending on patient’s cooperation and the surgeon’s experience and expertise, the most common settings include a slit lamp, a minor procedure room with a supine patient, or an operating room, with the latter two settings relying on the use of an operating microscope. When obtaining a corneal biopsy, the visual axis should be maintained as much as possible, whilst the risk of perforation should be minimised. The favoured site for sampling in clinical practice is found at the leading edge of the lesion, as this site obtains both infected and non-infected tissues whilst avoiding necrotic tissue that is usually present in the central portion of the ulcer; biopsy of this necrotic zone could have a lower organism yield and a higher risk of perforation [87]. Corneal biopsy has also been used to help map the extent and margin of refractory AK before therapeutic keratoplasty to reduce and/or prevent the risk of reinfection of the donor’s corneal graft [95]. 

Some of the techniques through which corneal biopsies are obtained include freehand lamellar dissection using a metal or diamond blade (higher risk of perforation due to the freehand nature, along with increased risk of scarring and corneal irregularity during larger dissections), trephination (more precise with lower risk of perforation) [96], formation of a lamellar flap to perform biopsy on the inner-facing portion of the flap [97], suture pass (commonly performed using braided silk, though it does not allow histological analysis, as it does not yield a discrete tissue sample), and the use of a reverse-cutting blade [98]. Newer techniques include femtosecond laser-assisted corneal biopsy, in which a femtosecond laser is used to obtain a bladeless biopsy. Due to the precise nature of the laser, this technique is safer, having a much lower risk of perforation (especially with deeper infiltrates), as well as a lower risk of scarring-induced refractive changes due to the smoother corneal surface post-procedure in comparison to other techniques. It is hindered by the inferior cost-effectiveness and the theoretical risk of laser-induced tissue sterilisation at the biopsy margins [99,100]. Another emerging technique is anterior segment optical coherence tomography (AS-OCT)-assisted corneal biopsy. Here, AS-OCT is used in real time to localise the infiltration and control the depth and extent of dissection, increasing accuracy and decreasing the risk p perforation [101].

## 4. Polymerase Chain Reaction

In recent years, the use of molecular diagnostic tests has become increasingly popular for the diagnosis of infectious keratitis, including AK [61,67,102,103,104,105]. Polymerase chain reaction (PCR) is a rapid and highly sensitive enzymatic assay in which a targeted deoxyribonucleic acid (DNA) fragment can be amplified and detected within a DNA sample. The process involves repeated cycles of denaturation, annealing, elongation and replication of the target DNA sequence, generating billions of copies of this sequence. As such, only a small amount of DNA is required to yield a positive result. PCR requires a number of key players, including template DNA; pre-determined short and single-stranded DNA sequences that complement the targeted DNA, which are known as primers; nucleotides; and DNA polymerase [106]. 

The most common DNA target sequence used for *Acanthamoeba* in PCR is the 18S ribosomal DNA (rDNA) sequence, which encodes for 18S rRNA, which is the main component of the 40S small subunit [107,108,109,110,111,112]. Studies show that PCR is not only a more sensitive diagnostic test than culture, but it is also much faster, having a turnaround rate of a few hours [64,66,107,113,114]. However, similar to culture, single PCR assays can misdiagnose AK due to having a sensitivity of around 73.3% [64], though the use of two or more different primers increases its sensitivity [66,114], having a reported sensitivity of 93.3–100% and a specificity of 99.3–100% for combined PCR assays [64,113], highlighting the superiority of combined assays over a single assay. Furthermore, due to a reported high negative predictive value (99.7–100%) for combined PCR assays [64], the negative test is more significant than it is for negative culture.

Echoing the problems encountered during the culture of *Acanthamoeba*, chemicals and dyes used in the diagnosis and treatment of eye disease can lead to false negative results due to having an inhibitory effect on PCR. Affected dyes include fluorescein, lissamine green dyes, rose Bengal dyes and oxybuprocaine drops [111,115,116,117]. Additionally, PCR can be inhibited by endogenous inhibitors found in aqueous and vitreous fluids [118]. To address this issue, sampling should ideally be performed prior to any specific treatment. Besides PCR of corneal epithelial samples, tear sample PCR [66] and PCR of CLs, CL cases or solutions [107] might also be helpful in improving the diagnosis of AK.

Recently, real-time or quantitative PCR has gained increased popularity. It allows the analysis of targeted DNA in real time by monitoring fluorescence levels [119]. This approach significantly shortens the turnover to times as short as 3.5 h [110,119,120,121,122]. It was first used to detect *Acanthamoeba* 18S ribosomal DNA (rDNA) by Rivière et al. in 2006 [123]. The limitation of this study was that primers and probes were designed against only six *Acanthamoeba* T4 genotype DNA sequences, without any validation using clinical samples. Subsequently, Qvarstrom et al. used a triplex real-time PCR assay (to detect Balamuthia mandrillaris, Naegleria fowleri, and *Acanthamoeba* species), with the primers and probe for *Acanthamoeba* designed against 40 different *Acanthamoeba* 18S rRNA sequences and tested against seven strains extracted from four genotypes (T1, T4, T7, and T10) [110].

Since this study was carried out, a number of studies have reported an increased preference for real-time PCR over conventional PCR in clinical laboratories [124]. The major advantage of real-time PCR is that there is no need for post-amplification handling, leading to faster analysis and reduced risk of amplicon contamination, whilst also providing an estimate of pathogen load [121,125,126]. 

Other modern PCR techniques that have been used in the diagnosis of AK include loop-mediated isothermal amplification (LAMP), nested PCR and nanoparticle-assisted PCR (nanoPCR). In LAMP, the target DNA sequences are amplified under isothermal conditions (temperature ranges from 58 to 65 °C). This technique has been shown to have efficacy comparable to that of conventional PCR in AK diagnosis whilst being less time consuming due to the lack of a thermal cycler [127,128]. In nested PCR, two sets of primers and two successive PCR reactions are used to improve the sensitivity and specificity of diagnosis [129]. Finally, in nanoPCR, nanomaterials are introduced into the PCR mixture to enhance PCR thermal conductivity, addressing the heat transfer limitation, which is known to reduce PCR efficiency in conventional PCR thermal cyclers, thus increasing target DNA yield [130,131,132]. 

It is clear that the high sensitivity and specificity of modern PCR techniques could be a game changer in the diagnosis of AK, albeit in combination with other techniques. However, it is worth noting that PCR is only able to amplify a targeted DNA sequence based on a specific primer. It, thus, only analyses the target organism and does not have the ability to detect other organisms, unless multiplex PCR is used. Multiplex PCR enables simultaneous amplification of multiple different DNA sequences, allowing the detection of different organisms via one analysis [110]. However, clinical suspicion of the organisms involved is required to decide which target DNA sequences are to be analysed, which limits its current usability. 

## 5. In Vivo Confocal Microscopy

In vivo confocal microscopy (IVCM) has emerged as a reliable corneal imaging tool that is employed to diagnose infectious keratitis, including fungal keratitis and AK [46,58,61,67,133,134,135]. It allows the high-resolution in vivo evaluation of corneal structures and pathologies at cellular and sub-cellular levels in a non-invasive manner [136,137,138]. It is considered to be a non-invasive form of corneal biopsy, and it works either by using the light reflected from within the illuminated tissue during changes in the refraction index or using fluorescent agents to aid recognition of inter- and intra-cellular details. A light beam is passed through a light source aperture and then focused via an objective lens into a small focal volume within the cornea. A mixture of emitted, as well as reflected, light from the illuminated spot is then recollected using the objective lens, and a beam splitter separates the light mixture and reflects the light into the detection apparatus. After passing through a pinhole, the light is detected via a photodetection device, which transforms the light signal into an electrical signal, which is recorded. The detector aperture obstructs the light that does not come from the focal point, resulting in sharper images, thus optimising illumination and detection for only a single spot, in contrast to conventional light microscopy. 

The principle of confocal microscopy was first described in 1957 by Minsky [139], though since that year, a number of different types of IVCM have been developed and applied to clinical practice (Table 1). An example is tandem scanning IVCM, which was developed by Petráň and Hadravský based on Nipkow’s work, which produces a real-time image that can be directly viewed [140]. Another type of IVCM is scanning slit IVCM, in which the scanning time is markedly reduced through the use of an adjustable slit in which all points along its axis are simultaneously scanned. This type of IVCM is able to produce real-time, high contrast, single-video-frame images of 2-micrometRE sections through the full corneal thickness [141]. It is used in practice (ConfoScan, Nidek Technologies) [133] and has lateral and axial resolutions of 1 μm and up to 24 μm, respectively [142]. The other IVCM technique that is in clinical use is laser scanning IVCM, which was developed by Webb [143,144,145]. During this process, a coherent laser light source is used, with the laser beam being scanned via a set of galvanometer scanning mirrors about the XY plane. The reflected light is then refocused via the microscope objective lens, via by the scanning mirrors and imaged onto a pinhole aperture located in front of the photomultiplier. The Heidelberg Retina Tomograph (HRT) (Heidelberg Engineering, Heidelberg, Germany), combined with a specially designed and mountable objective system, which is named the ‘Rostock Cornea Module’ (RCM) [146], is the main example, and it seems to be one of the most commonly used IVCMs in clinical practice [67,147]. It operates by scanning a 670-nanometre red wavelength diode laser beam in a raster pattern over the field of view [148], and it is able to produce high-resolution images, having a lateral resolution of 1 μm, an axial resolution of 7–8 μm (depending on the contact cap used) and a 400× magnification [142,149,150,151].

In AK, IVCM has been shown to be a valuable diagnostic modality. Compared to culture and PCR, IVCM exhibits a superior diagnostic performance, having an overall sensitivity of 77–100% and specificity of 84–100% [65,67,134,158,159,160]. There are various presented morphological features of *Acanthamoeba* on IVCM, which depend on the stage of the disease’s process (Figure 2). These features include double-walled cysts (dormant form) with a 15–30-micrometre diameter located at the epithelium and/or stroma (the most common reported IVCM AK feature), a 25–40-micrometre hyper-reflective trophozoites (active form) bright spots and signet rings and a perineural hyper-reflective patchy infiltrate with surrounding hyper-reflective spindle-shaped materials [133,149,161,162,163,164,165,166]. 

Another feature of AK via IVCM is loss of the normal keratocyte morphology in the anterior stroma [167], which would usually consist of bright ovoid nuclei and barely visible cellular processes [168]. However, after injury, the keratocytes reduce production of molecules that contribute to cellular transparency, increasing the visibility of their cellular processes via IVCM [169,170]. Interestingly, following the use of topical steroids, cysts have been observed to form clusters [167,171], which were associated with a poor prognosis [172]. The reason that this cyst-clustering occurs is currently unknown, but it has been postulated to resemble the mechanism of “biofilm formation” observed in other types of infectious keratitis [58]. Occasionally, IVCM might also show co-infection by other organisms, such as fungi [46].

There are a number of clinical scenarios of AK in which IVCM would be useful. Firstly, since AK progresses rapidly into the deep stroma, as mentioned above, cultures from corneal scrapes would not be able to pick it up. In fact, it is thought that culture-positive AK is only part of the problem, as IVCM evidence of AK is 10 times the culture-positivity rate [114]. Therefore, IVCM would be of great utility in culture-negative cases and in cases of deep infiltrates not being accessible to corneal scrapings and necessitating corneal biopsy, as it provides a non-invasive alternative [134]. Secondly, IVCM might also be essential in cases in which anti-*Acanthamoeba* therapy would have already been initiated, as this would render the anterior stroma sterile, whilst amoebae persist in the deep stroma. Furthermore, as mentioned above, some anti-*Acanthamoeba* drugs, along with common eye drops, can interfere with culture and PCR analysis [69,70,111,115,116,117,118], making IVCM a great option. It could also be useful in cases of AK (or other causes of infectious keratitis) post-corneal surgery, such as intracorneal rings, LASIK, radial keratotomy and other incisional refractive surgeries, as the infection would be deep seated and preclude the use of standard microbiological investigations. Furthermore, it also has a role in monitoring the response to treatment, as it serves as a pre-transplant screening tool for any residual cysts.

Unfortunately, even though IVCM represents a rapid and non-invasive technique with excellent diagnostic performance in AK, it is still not widely available and affordable. Furthermore, the interpretation of images entails a steep learning curve, with the diagnostic accuracy improving with the increase in the experience of the IVCM grader [173]. However, the rapid nature of IVCM means that diagnostic delays would be minimised, leading to timely administration of appropriate treatment and improved prognosis.

## 6. Anterior Segment Optical Coherence Tomography

Recently, AS-OCT scanning has been increasingly used in the diagnosis of infectious keratitis, including AK. It uses low-coherence interferometry to enable non-contact and non-invasive detailed imaging of the layers of the cornea, whilst accurately delineating the depth and extent of corneal ulceration, infiltration and haze, in order to characterize, quantify and monitor the progress of a variety of corneal pathologies [174,175,176,177,178,179].

The use of OCT for anterior segment imaging was first reported in 1993 [180], which was soon after it was initially used for in vivo imaging of the retina [181] OCT generates two- or three-dimensional tomographic images by measuring the echo time delay of light backscattered from tissue structures. Light from a low-coherence light source is split into two paths using a beam splitter and directed into the two arms of an interferometer. In the reference arm, the light is reflected by a mirror, whilst in the sample arm, the light is back-scattered by the tissues. Illumination properties of the sample beam, such as shape, depth of focus and intensity distribution, are determined via optical components in the sample arm. Since in tissues different structures have different refractive indices, light is back-scattered at interfaces between layers of different refractive indexes. The light returning from the reference and sample arms is then recombined at the beam splitter and guided towards the detector. Interferences occur only when the pathway between the two beams is located within the coherence length of the light source. 

To obtain depth resolution, time-domain (TD) or Fourier-domain (FD) techniques are used. In TD-OCT, a scanning reference delay is utilised whilst translating the reference arm mirror. The light is collected using a point detector, and the reference mirror is moved at constant velocity, with a complete travel of this mirror being called an A-scan, in an analogy to ultrasound technology. At different positions of the reference mirror, different structures within the sample lead to interferences, leading the sequential detection of light echoes. The most common available TD-OCT devices are the Visante OCT (Carl Zeiss Meditec, Oberkochen, Germany) and the Heidelberg slit lamp OCT (SL-OCT, Heidelberg Engineering GmbH, Heidelberg, Germany) [182]. The Visante OCT has a wavelength of 1310 nm, a 16-millimetre scan width, a 6-millimetre scan depth, an axial resolution of 18 μm and a transverse resolution of 60 μm; the Heidelberg SL-OCT has a similar wavelength (1310 nm), scan width (15 mm) and depth (7 mm), though it has an axial resolution of >25 μm and transverse resolution of 20–100 μm [183,184,185]. However, a major drawback of both these TD-OCTs is the low A-scan rate (2000 A-scans per second using the Visante OCT and 200 A-scans per second using the Heidelberg SL-OCT) and the lower sensitivity than FD-OCT [186].

On the other hand, in FD-OCT, modulations in the source spectrum are collected, with all spectral components being captured simultaneously through the use of a spectrometer, which spectrally analyses light. In this experiment, the mirror in the reference arm is static, and a full image is achieved via one camera shot. The wave number-dependent signal is transformed into the axial scan information using an inverse Fourier transform. The main type of FD-OCT in use in ophthalmology is spectral-domain OCT (SD-OCT), with the main examples being Spectralis (Heidelberg Engineering GmbH, Heidelberg, Germany), RTVue (Optovue, Inc., Fremont, CA, USA) and its smaller counterpart iVue80 and Cirrus OCT (Carl Zeiss Meditec). These SD-OCT techniques have acquisition speeds 10 to 100 times that of TD-OCT [183,187]. For example, RTVue captures 26,000 A-scans per second, and it detects signals from the entire depth in parallel rather than serially, making it much more efficient and faster without losing its signal-to-noise ratio. It also has a depth resolution that is over three times higher (5 μm) than that of the Visante [187]. However, the horizontal scan width and scan depth of SD-OCT are shorter than those of TD-OCT devices [188].

Another FD-OCT technology is based on swept-source (SS) technology, with the light source wavelength being rapidly swept within the laser, allowing the use of a point detector and high acquisition rates of up to several MHz [176,179,189,190]. An example is Casia SS-1000 OCT (Tomey, Nagoya, Japan), which is specifically designed to be an AS-OCT. It uses a 1310-nanometre wavelength, has a horizontal scan width of 16 mm and carries an axial resolution of 10 μm and a transverse resolution of 30 μm, as well as a high scanning speed of 30,000 A-scans per second [191].

Recently, ultra-high-resolution OCT (UHR-OCT) has been developed, having reported axial resolution of 1–4 μm and a scan width of 5–12 mm [188,192,193,194]. This resolution is made possible via the utilisation of a light source with a broad bandwidth of more than 100 μm and a spectrometer that is able to detect the fringes reflected from both of the reference and sample arms. Commercially available UHR-OCT machines include Bioptigen Envisu (Bioptigen Inc., Research Triangle Park, NC, USA) and the SOCT Copernicus HR (Optopol Technologies SA, Zawiercie, Poland) [195]. The technical specifications of some of the commercially available AS-OCT machines are summarised in Table 2.

In AK, AS-OCT has been shown to provide detailed visual images of radial keratoneuritis and perineural infiltrates, which appear as hyper-reflective bands in the corneal stroma of varying widths (20–200 μm) and depths (subepithelial to mid-stroma) [199,200], as shown in Figure 3. However, the sensitivity and specificity of AS-OCT in AK diagnosis are not yet known. Unfortunately, unlike IVCM, at the moment, AS-OCT is not able to identify cysts or trophozoites because of its limited resolution power. However, with constant improvements in OCT technology, this process might be a possibility in the future, which would potentially transform the whole diagnostic process of AK. At present, its use in AK is, therefore, limited to being an adjunct to be used alongside other diagnostic techniques. 

## 7. Conclusions and Future Directions

The diagnosis of AK remains challenging. Early diagnosis of AK is essential to ensure prompt treatment and good clinical outcomes. Traditionally, conventional culture and microscopy were considered to be the gold standard for diagnosing AK, but emerging evidence regarding PCR and IVCM have supported their diagnostic utility in treating AK in the clinic.

There has also been an increased interest in using AI-based models to assist the diagnosis of infectious keratitis, including AK [201]. Most AI research has focused on deep learning (DL), where convolutional neural networks (CNN) are used to develop sophisticated AI systems that are able to analyse high-definition data, such as images, in a quick and accurate manner [202]. Within image processing applications, CNNs have multiple convolutional layers that capture high-level features of the image into multiple abstraction levels for processing, improving accuracy and versatility [203]. The emergence of DL and CNN has transformed the way that we perform medical data analysis and medical image recognition, and in ophthalmology, most DL-based AI systems have been developed for screening and diagnosis of posterior segment pathology, such as glaucoma, macular degeneration, diabetic retinopathy, retinal breaks and retinal detachment, using fundus images, OCT and/or visual fields [204,205,206,207,208,209,210,211]. Recently, studies have successfully applied AI to slit lamp images for the diagnosis of infectious keratitis and shown improved diagnostic performance and accuracy, potentially involving fewer diagnostic delays [212,213,214,215,216]. These AI systems have also been applied to IVCM for the diagnosis of infectious keratitis, having a reported sensitivity of 91.9% and specificity of 98.3% in diagnosing fungal keratitis [217,218]. With regard to AK, studies report a sensitivity and specificity of 91.4% and 98.3%, respectively, for DL-based IVCM analysis [219], as well as a diagnostic accuracy of 83.81–97.9% via slit lamp image analysis [220,221]. These AI systems reduce the need for technical expertise to be applied to interpreting complex images whilst also decreasing clinicians’ workloads. At the moment, these revolutionary advances are being treated as adjuncts in diagnostic pathways. However, with further development and research, they have the potential to completely transform the diagnostic landscape. 

Another emerging diagnostic tool that has demonstrated promise is next-generation sequencing (NGS), which enables massive parallel sequencing of DNA and RNA to detect different organisms. The ability to rapidly detect multiple organisms from cornea samples could further reduce the reliance on a clinician’s subjective experience and clinical judgement [222,223,224,225]. Early studies have shown that NGS is able to detect *Acanthamoeba* [226,227], along with providing valuable information about genotypes and co-infecting organisms. However, it has a lower sensitivity (88%) than real-time PCR for the detection of *Acanthamoeba*-specific DNA (specificity 100%) [227]. Further research and technique development is required to assess the role of NGS in diagnosing AK.

As the treatment outcome of AK is highly time sensitive, we need to maintain a low threshold of suspicion when assessing patients with possible AK, particularly those who present with a history of CL wear and/or trauma. Microbiological findings should be clinically correlated, and a combination of diagnostic techniques provides the best chance of ensuring timely diagnosis and prompt treatment of AK, which would, ultimately, improve the visual prognosis.

## Figures and Tables

**Figure 1 diagnostics-13-02655-f001:**
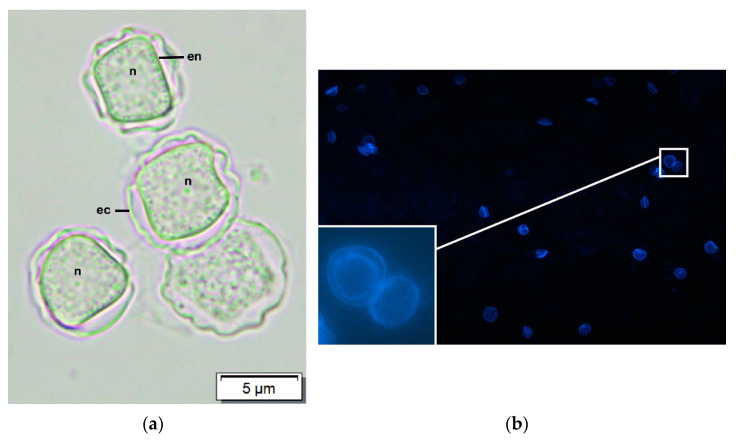
Direct microscopy and staining (**a**) *Acanthamoeba* cysts viewed the under a light microscope, which shows nuclei (n), double-layered walls, polygonal inner walls (endocyst, en) and thick wrinkled outer walls (ectocyst, ec) (reproduced in an unchanged format under the terms of the CC BY-NC 4.0 license, http://creativecommons.org/licenses/by-nc/4.0, accessed on 17 July 2023) [77]; (**b**) double-walled *Acanthamoeba* cysts stained with Calcofluor white (reproduced in an unchanged format under the terms of the CC BY 4.0 license, https://creativecommons.org/licenses/by/4.0/, accessed on 17 July 2023) [78].

**Figure 2 diagnostics-13-02655-f002:**
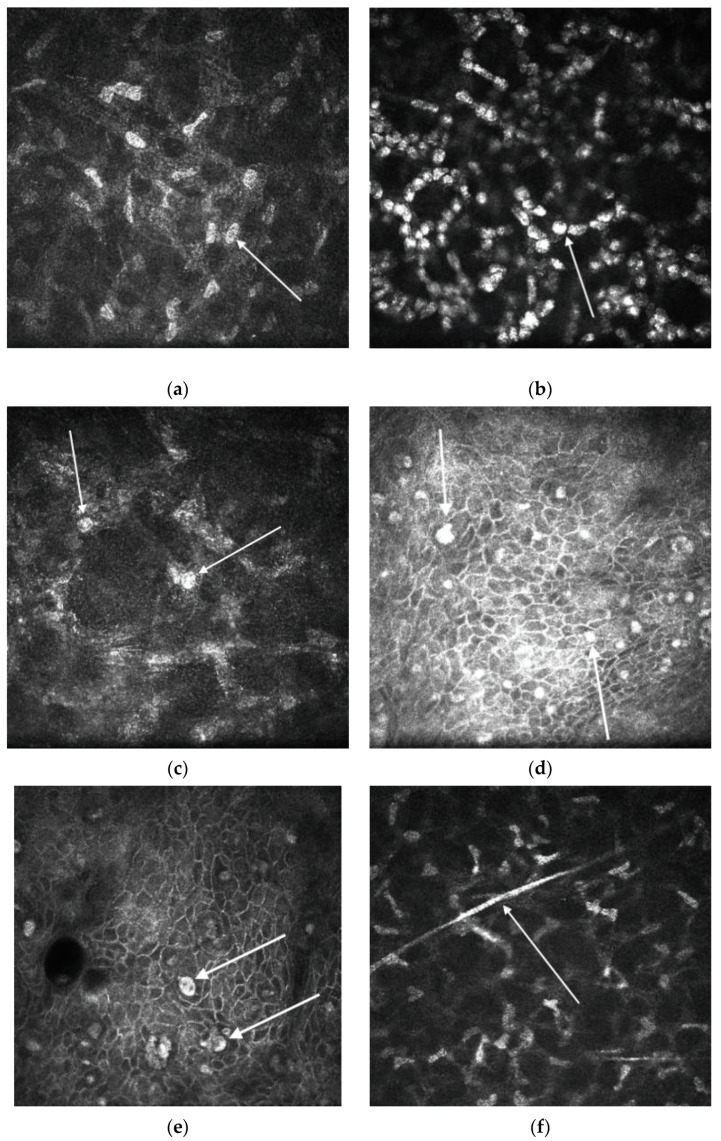
Features of *Acanthamoeba* keratitis subjected to in vivo confocal microscopy, which were obtained using Heidelberg Retina Tomograph 3 and Rostock Cornea Module (HRT3-RCM); (**a**) normal keratocyte morphology appearance (bright ovoid nuclei and barely visible cellular processes) with (**b**) Hyper-reflective trophozoites, (**c**) double-walled cysts, (**d**) bright spots, (**e**) signet rings (**f**) perineural hyper-reflective patchy infiltrate.

**Figure 3 diagnostics-13-02655-f003:**
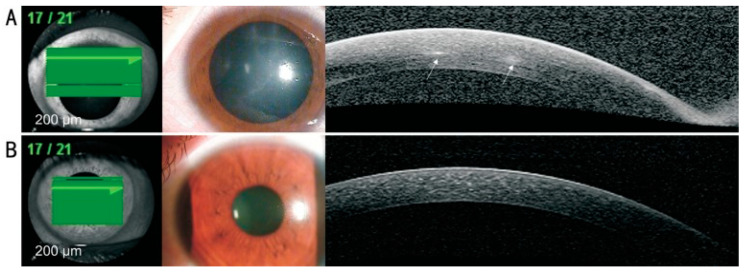
Anterior segment OCT showing reflective bands in the corneal stroma that correspond to the area of radial keratoneuritis (**A**), with complete resolution after treatment for Acanthamoeba keratitis (**B**) (reproduced unchanged under the terms of the CC BY-NC-ND 4.0 license, https://creativecommons.org/licenses/by-nc-nd/4.0/, accessed on 17 July 2023) [200].

**Table 1 diagnostics-13-02655-t001:** Features of various in vivo confocal microscopy (IVCM) techniques.

Characteristic	Tandem Scanning IVCM *	Scanning Slit IVCM	Laser Scanning IVCM
Main example	Tandem Scanning Confocal Microscope Tandem Scanning Corporation (Reston, Va, USA)	ConfoScan 4 Nidek Technologies (Gamagori, Japan/Padova, Italy)	Heidelberg retina tomograph and Rostock cornea module (HRT-RCM) Heidelberg Engineering (Heidelberg, Germany)
Light source	Mercury and Xenon [152]	Halogen [148,152,153]	Diode laser [148,149]
Light source wavelength	400–700 nm [152]	370–510 nm [152]	670 nm [148]
Illumination and light detection	Rotating Nipkow disk (64,000 holes of 20–60 microns in diameter) [140,152]	Two conjugate slits [149,152]	Two scanning mirrors and one scanner [152]
Lateral resolution	N/A	1 μm [137]	1 μm [151]
Axial resolution	9 μm [150,154]	24 μm [150]	7–8 μm [142,150]
Magnification	60× ** [155]	500× [156]	400× [149]
Advantages	Diffraction-limited resolution, optical sectioning, faster than LSCM [157]	Faster scanning [157]	Diffraction-limited resolution, versatile, optical sectioning [157]
Disadvantages	Pinhole cross-talk, artefacts from disc–camera synchronisation, fixed pinhole size, phototoxicity [157]	Lower resolution [157]	Phototoxicity, slow speed, axial resolution at depth [157]

* No longer in widespread clinical use. ** When used to observe the retina; corneal data not available.

**Table 2 diagnostics-13-02655-t002:** Comparison between some of the anterior segments using commercially available OCT machines.

Characteristic	Time-Domain OCT	Fourier-Domain OCT		Ultra-High-Resolution OCT
		Spectral-Domain OCT	Swept-Source OCT	
Examples in clinical use	1. Visante OCT (Carl Zeiss Meditec, Jena, Germany) 2. Heidelberg slit lamp OCT (Heidelberg Engineering, Heidelberg, Germany)	Spectralis (Heidelberg Engineering, Heidelberg, Germany) 2. iVue80 (Optovue, Inc., Fremont, CA, USA) 3. Cirrus OCT (Carl Zeiss Meditec, Jena, Germany)	1. Casia SS-1000 OCT (Tomey, Nagoya, Japan) 2. Triton OCT (Topcon Corporation, Tokyo, Japan)	1. SOCT Copernicus HR (Optopol Technologies SA, Zawiercie, Poland)
Optical source	Superluminescent diode [176]	Superluminescent diode [176]	Swept-source laser [176]	Superluminescent diode [176]
Wavelength	1 = 1310 2 = 1310 nm [183,184,185]	1 = 820 nm 2 = 840 nm 3 = 840 nm [176]	1 = 1310 nm 2 = 1310 nm [176]	1 = 850 nm [196]
Scan width	1 = 16 mm 2 = 15 mm [183,184,185]	1 = 6 mm 2 = 13 mm 3 = 6 mm [176]	1 = 16 mm 2 = 16 mm [176,197]	1 = 10 mm [196]
Scan depth	1 = 6 mm 2 = 7 mm [183,184,185]	1 = 2 mm 2 = 2–2.3 mm (retina) 3 = 2 mm [176,198]	1 = 6 mm 2 = 6 mm [176]	N/A **
Axial resolution *	1 = 18 μm 2 = >25 μm [183,184,185]	1 = 7 μm 2 = 5 μm 3 = 5 μm [176,198]	1 = 10 μm 2 = 8 μm [176]	1 = 3 μm [196]
Transverse resolution *	1 = 60 μm 2 = 20–100 μm [183,184,185]	1 = 20 μm 2 = 15 μm 3 = 15 μm [176,198]	1 = 30 μm 2 = 30 μm [176]	1 = 12–18 μm [196]
A-scan rate	1 = 2000 scans/s 2 = 200 scans/s [176,186]	1 = 40,000 scans/s 2 = 80,000 scans/s 3 = 27,000 scans/s [176,198]	1 = 30,000 scans/s 2 = 100,000 scans/s [176]	1 = 52,000 scans/s [196]

* Some measurements are specific to the posterior segment used and might vary for anterior segment OCT; ** not documented in the literature.

## Data Availability

This review has no new research data.

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
