# Peer review of "Diagnosis of *Acanthamoeba* Keratitis: Past, Present and Future"

_diagnostics, 2023, doi:10.3390/diagnostics13162655_

Round 1
Reviewer 1 Report
In this review paper, Azzopardi aims to present a comprehensive overview of the diagnosis of Acanthamoeba keratitis (AK). The review provides a thorough examination of the current diagnostic methods used for AK, including the historical use of corneal scraping for microbiological culture as the gold standard. The author acknowledges the method's technical ease, accessibility, and cost-effectiveness but also highlights its limitations, such as the long diagnostic turnaround time and variable sensitivity, rendering it inadequate as a sole diagnostic test in clinical practice.
1. While the review exhibits commendable writing in various aspects, it falls short in providing sufficient substance and depth in its discussions. It heavily relies on findings from prior review papers, lacking novel insights. Additionally, certain pertinent clinical topics are noticeably absent, such as the critical matter of using corticosteroids to alleviate the intense inflammation associated with Acanthamoeba keratitis. This omission is particularly relevant for clinicians and warrants further attention.
2. Regarding the section on Anterior Segment Optical Coherence Tomography (AS-OCT), some suggestions for improvement can be made. Expanding on the applications of AS-OCT in infectious keratitis and AK, with specific examples of its clinical utility and impact on patient care, would enhance the section's value. Additionally, including citations for the mentioned studies and sources to support claims about the advantages and limitations of different OCT techniques will increase the section's credibility and reliability.
3. The "Conclusions and future directions" section effectively summarizes the current status of AK diagnosis, providing valuable insights into promising technologies like AI and NGS. The section also offers practical recommendations for clinicians to improve AK management. As an editor, I find this section to be well-written, insightful, and a valuable addition to the manuscript.
Overall, the review has valuable contributions to the field of Acanthamoeba keratitis diagnosis, but certain areas require improvement to ensure thoroughness and accuracy. Addressing the mentioned suggestions will significantly enhance the manuscript's overall quality and impact.
Author Response
All of the reviewer comments have been included chronologically, with authors’ answers in bold.
In this review paper, Azzopardi aims to present a comprehensive overview of the diagnosis of Acanthamoeba keratitis (AK). The review provides a thorough examination of the current diagnostic methods used for AK, including the historical use of corneal scraping for microbiological culture as the gold standard. The author acknowledges the method's technical ease, accessibility, and cost-effectiveness but also highlights its limitations, such as the long diagnostic turnaround time and variable sensitivity, rendering it inadequate as a sole diagnostic test in clinical practice.
While the review exhibits commendable writing in various aspects, it falls short in providing sufficient substance and depth in its discussions. It heavily relies on findings from prior review papers, lacking novel insights. Additionally, certain pertinent clinical topics are noticeably absent, such as the critical matter of using corticosteroids to alleviate the intense inflammation associated with Acanthamoeba keratitis. This omission is particularly relevant for clinicians and warrants further attention.
Authors’ comment:
We would like to thank the reviewer for this comment. Since this narrative review focused on the Diagnostic aspect of Acanthamoeba keratitis, clinical topics such as use of corticosteroids to alleviate AK-associated inflammation are beyond the scope of this review.
Regarding the section on Anterior Segment Optical Coherence Tomography (AS-OCT), some suggestions for improvement can be made. Expanding on the applications of AS-OCT in infectious keratitis and AK, with specific examples of its clinical utility and impact on patient care, would enhance the section's value. Additionally, including citations for the mentioned studies and sources to support claims about the advantages and limitations of different OCT techniques will increase the section's credibility and reliability.
Authors’ comment:
We would like to thank the reviewer for this comment. As the reviewer has mentioned, AS-OCT has a wide range of applications apart from in AK diagnostics, including in other forms of infective keratitis, corneal dystrophies, corneal ectasias, traumatic corneal injuries, dry eye disease, corneal transplantation (planning pre-transplant and monitoring post-transplant), and refractive surgery. It has also been shown to be useful in the diagnostics for glaucoma. However, since this review focuses on the diagnostic options in AK, this is beyond the scope of the review. Furthermore, when possible we have cited all studies and sources that could support the advantages and limitations of each diagnostic technique. However, at times we also relied on the clinical expertise and experience of the authors as ophthalmologists.
The "Conclusions and future directions" section effectively summarizes the current status of AK diagnosis, providing valuable insights into promising technologies like AI and NGS. The section also offers practical recommendations for clinicians to improve AK management. As an editor, I find this section to be well-written, insightful, and a valuable addition to the manuscript.
Authors’ comment:
We would like to thank the reviewer for this encouraging comment.
Overall, the review has valuable contributions to the field of Acanthamoeba keratitis diagnosis, but certain areas require improvement to ensure thoroughness and accuracy. Addressing the mentioned suggestions will significantly enhance the manuscript's overall quality and impact.
Reviewer 2 Report
For Authors
The review manuscript is well-written.
I have a comment the authors need to address.
Figure 3 picture showed anterior segment-OCT showing radial keratoneuritis.
But I don't think this picture is real thing. Because corneal thickness is normal, the cornea doesn't show edematous, the picture doesn't show keratitis.
My recommendation is to show anterior segment photo together in this part.
Author Response
All of the reviewer comments have been included chronologically, with authors’ answers in bold.
The review manuscript is well-written.
Have a comment the authors need to address:
Figure 3 picture showed anterior segment-OCT showing radial keratoneuritis.
But I don't think this picture is real thing. Because corneal thickness is normal, the cornea doesn't show edematous, the picture doesn't show keratitis.
My recommendation is to show anterior segment photo together in this part.
Authors’ comment:
We would like to thank the reviewer for their comments. The original Figure 3 showed radial keratoneuritis (on AS-OCT) in an early AK. This photo is the authors’ own property, and the patient involved had a confirmed diagnosis of AK. However, we have changed the Figure to one with more obvious radial keratoneuritis and corneal oedema (obtained from another study published in 2018, reproduced unchanged under the terms of the CC BY-NC-ND 4.0 license, https://creativecommons.org/licenses/by-nc-nd/4.0/).
Reviewer 3 Report
Dear Authors,
I wish to submit my review of the article titled: "Diagnosis of Acanthamoeba Keratitis: Past, Present and Future"
The Authors narratively describe and review the current diagnostic tools for Acanthamoeba keratitis (AK) and should be commended for their work.
However, the manuscript may benefit from a minor review.
Indeed, the manuscript would benefit from a paragraph describing the "clinical signs" and Important clinical characteristics for the differential diagnosis of AK compared to keratitis due to other infectious agents.
Moreover, adding a diagnostic algorithm overview (Figure) may help the reader to use the manuscript's information pragmatically.
Line 392, There is a possible typo mistake. The number 9 is placed after the dot. Is it a reference?
Author Response
In this section, all of the reviewer comments have been included chronologically, with authors’ answers in bold.
Dear Authors,
I wish to submit my review of the article titled: "Diagnosis of Acanthamoeba Keratitis: Past, Present and Future"
The Authors narratively describe and review the current diagnostic tools for Acanthamoeba keratitis (AK) and should be commended for their work. However, the manuscript may benefit from a minor review. Indeed, the manuscript would benefit from a paragraph describing the "clinical signs" and Important clinical characteristics for the differential diagnosis of AK compared to keratitis due to other infectious agents.
Authors’ comment:
We would like to thank the reviewer for this comment. We have added this information to the Introduction section.
Moreover, adding a diagnostic algorithm overview (Figure) may help the reader to use the manuscript's information pragmatically.
Authors’ comment:
The authors feel that once a diagnosis of Acanthamoeba keratitis (AK) is suspected, depending on the clinical picture and the likelihood of diagnosis, as many of the available investigations should be performed as early as possible. This is not only due to the low sensitivity and specificity of diagnostic techniques when performed in isolation, but also due to the high sensitivity and specificity of diagnostic investigations when performed together, and the sight-threatening, progressive nature of AK (as detailed in our review). Furthermore, the specific combination of diagnostic techniques employed in any clinical scenario would not only depend on the clinical picture and the likelihood of AK, but also on the diagnostic armamentarium available to different healthcare systems. Therefore, a rigid diagnostic algorithm is difficult to ascertain and would seem presumptuous. It would be more beneficial if this is done as a working group consensus with global KOL, and this is beyond the scope of this review.
Line 392, There is a possible typo mistake. The number 9 is placed after the dot. Is it a reference?
Authors’ comment:
This is indeed a typo and has been removed from the text.
Round 2
Reviewer 1 Report
The authors have appropriately dealt with the majority of my concerns, resulting in no further questions on my part
Reviewer 3 Report
Dear Authors,
I wish to submit the review for the article titled: "Diagnosis of Acanthamoeba Keratitis: Past, Present and Future"
The Authors provided satisfactory answers to the questions that I have previously raised. Accordingly, They amended and expanded the manuscript